# Systemically Identifying Triple-Negative Breast Cancer Subtype-Specific Prognosis Signatures, Based on Single-Cell RNA-Seq Data

**DOI:** 10.3390/cells12030367

**Published:** 2023-01-19

**Authors:** Kaiyuan Xing, Bo Zhang, Zixuan Wang, Yanru Zhang, Tengyue Chai, Jingkai Geng, Xuexue Qin, Xi Steven Chen, Xinxin Zhang, Chaohan Xu

**Affiliations:** 1College of Bioinformatics Science and Technology, Harbin Medical University, Harbin 150081, China; 2Department of Pharmacology, State-Province Key Laboratories of Biomedicine-Pharmaceutics of China, Key Laboratory of Cardiovascular Medicine Research, Ministry of Education, College of Pharmacy, Harbin Medical University, Harbin 150081, China; 3Department of Public Health Sciences, Division of Biostatistics, University of Miami Miller School of Medicine, Miami, FL 33136, USA; 4Sylvester Comprehensive Cancer Center, University of Miami Miller School of Medicine, Miami, FL 33136, USA

**Keywords:** single-cell RNA-seq, TNBC subtype-specific, prognosis signature

## Abstract

Triple-negative breast cancer (TNBC) is a highly heterogeneous disease with different molecular subtypes. Although progress has been made, the identification of TNBC subtype-associated biomarkers is still hindered by traditional RNA-seq or array technologies, since bulk data detected by them usually have some non-disease tissue samples, or they are confined to measure the averaged properties of whole tissues. To overcome these constraints and discover TNBC subtype-specific prognosis signatures (TSPSigs), we proposed a single-cell RNA-seq-based bioinformatics approach for identifying TSPSigs. Notably, the TSPSigs we developed mostly were found to be disease-related and involved in cancer development through investigating their enrichment analysis results. In addition, the prognostic power of TSPSigs was successfully confirmed in four independent validation datasets. The multivariate analysis results showed that TSPSigs in two TNBC subtypes-BL1 and LAR, were two independent prognostic factors. Further, analysis results of the TNBC cell lines revealed that the TSPSigs expressions and drug sensitivities had significant associations. Based on the preceding data, we concluded that TSPSigs could be exploited as novel candidate prognostic markers for TNBC patients and applied to individualized treatment in the future.

## 1. Introduction

Triple-negative breast cancer (TNBC) is characterized by the loss of expression of the estrogen receptor (ER), progesterone receptor (PR), and the human epidermal growth factor receptor 2 (HER2) [1,2,3,4]. As the most aggressive and malignant subtype, it accounts for approximately 10–20% of all breast cancers (BCs) and has a poorer prognosis, when compared with other subtypes [5,6,7]. Since TNBC has significantly high heterogeneities in different molecular subtypes [8], some classical TNBC subtyping approaches have been proposed by using gene expression data and comprehensively applied in RNA-seq or microarray analysis. For example, the classical PAM50 method used for the BC subtyping was also applied in the TNBC subtyping, it contains five molecular subtypes (luminal A, luminal B, HER2-enriched, basal-like, and normal-like,) and subsequent studies found that the majority of TNBC tumors were of the basal-like PAM50 subtype [9,10,11]. Lehmann et al. first identified six molecular subtypes of TNBC, based on the microarray gene expression data from the bulk tumor tissues and named TNBCtype, which includes basal-like 1 (BL1), basal-like 2 (BL2), immunomodulatory (IM), luminal androgen receptor (LAR), mesenchymal (M), and mesenchymal stem-like (MSL) [12,13]. Subsequently, they found that transcripts in the IM and MSL subtypes were not contributed from cancer cells directly, so the classification criterion of TNBCtype was refined into TNBCtype-4 with BL1, BL2, LAR, and M [14]. These TNBC subtype classification methods not only provided the fundamental basis for the TNBC subtype stratification but also overall enhanced the personalized study in different TNBC subtypes. For example, Gulbahce et al. found significant differences in the PAM50 subtypes among different age groups in TNBC, and observed that older women with the basal-like TNBC subtype had worse survival outcomes [15]. Bareche et al. showed a significant genomic heterogeneity and distinct overall survival among the TNBCtype subtypes, providing new insights into the development of TNBC [16]. Harano et al. revealed that the immune cell infiltration in tumors varied in the different TNBCtype-4 subtypes, which might lead to different responses to chemotherapy [17]. Lehmann et al. conducted a comprehensive subtype-specific analysis of TNBC, they identified characteristics and treatment strategies of the TNBCtype-4 subtypes, which provided help for therapy of the TNBC subtypes [18].

Although TNBCtype-4 has been well established for TNBC research, the TNBC subtype-specific prognosis signatures have not been identified yet. The major reasons include the limited sample size of the gene expression data for each TNBC subtype and the heterogeneity of the different TNBC cohorts. The availability of the high-quality single-cell RNA-seq (scRNA-seq) of TNBC provides the opportunity to tackle this challenging problem.

Distinguishing to RNA-seq or microarray technologies, the later scRNA-seq technology enables to more accurately differentiate cell types at a single-cell resolution, which gives more promise for precisely characterizing the gene expression in individual tumor cells [19,20]. With the advantage of it, a couple of TNBC subtype researches have been conducted to improve the understanding of the TNBC subtypes. Karaayvaz et al. assigned 868 TNBC malignant epithelial cells to the TNBCtype-4 subtypes and found that multiple TNBC subtypes were expressed among the cells of each tumor, showing the intratumoral heterogeneity of the gene expression subtypes [21]. Zhou et al. used scRNA-seq data with 1189 cells to comprehensively analyze the gene regulatory network of the molecular subtypes in TNBC patients and dissected the critical genes for each molecular subtype, and found that the critical genes might play diverse roles in the different subtypes [22].

It’s critical to develop the TNBC subtype-specific prognosis gene signatures to facilitate the personalized treatment for TNBC. Here, we presented a single-cell RNA-seq-based method to identify the TNBC subtype-specific prognosis signatures. To comprehensively investigate and assess the identified TSPSigs in multiple aspects, the function enrichment, the differences from the background genes, prognosis, and a drug sensitivity analysis were carried out. We found that most TSPSigs were closely related to the development of TNBC and associated with drug sensitivities, and had a good prognostic efficiency, which could provide new insights into the exploration of the TNBC subtypes and would aid in the TNBC prognosis.

## 2. Materials and Methods

### 2.1. scRNA-Seq Datasets

We downloaded the TNBC scRNA-seq dataset and selected the GSE176078 cohort from the Gene Expression Omnibus (GEO) (https://www.ncbi.nlm.nih.gov/geo/ (accessed on 2 June 2021)), which comprised 42,512 cells of 10 TNBC patients and was detected by the Illumina NextSeq 500 platform (GPL18573) [23]. Considering the presence of the tumor microenvironment or non-tumor cells in GSE176078, we extracted 10,836 cancer epithelial cells following the original cell annotation file (annotated by Garnett v.0.1.4) of this dataset (Table 1). To avoid the influence of non-coding RNAs and pseudogenes annotated in the sequencing platform, we only retained 17,497 protein-coding genes according to the gene annotation file (GRCh38) from GEOCODE (https://www.gencodegenes.org/ (accessed on 8 December 2020)). At the same time, we removed low-quality cells and genes with a too low expression and removed all of the mitochondrial and ribosomal genes as well. Normalization by deconvolution was then performed for the unique molecular identifier (UMI) counts of GSE176078. The flowchart of this study was shown in Figure 1.

### 2.2. RNA-Seq and the Array Detection Datasets

Four external TNBC validation datasets and their corresponding clinical information were downloaded separately from TCGA (UCSC Xena, https://xenabrowser.net/ (accessed on 15 July 2021)) [24], METABRIC (cBioportal, http://www.cbioportal.org/ (accessed on 23 July 2021)) [25], and GEO. Among them, TCGA and METABRIC contain 123 and 299 TNBC patient samples, while GSE58812 [26] and GSE96058 [27] contain 107 and 151, respectively (Table 1).

### 2.3. TNBCtype-4 Subtypes Classification

To classify and identify the TNBC subtype for each cancer epithelial cell in GSE176078, the TNBCtype tool (http://cbc.mc.vanderbilt.edu/tnbc/ (accessed on 10 October 2021)) was used and the cells with the highest positive correlation coefficient with the TNBCtype-4 subtypes (BL1, BL2, LAR, and M) were respectively assigned to the corresponding TNBC subtypes. The same classification was performed in four validation datasets by the TNBCtype-4 subtyping approach.

### 2.4. TSPSigs Identification

Because the distinct TNBC subtype patients have a different clinical behavior [28,29], we thought that the recognized TSPSigs should be specific and different. Therefore, we identified TSPSig for each TNBC subtype using the CM1 score algorithm, a supervised univariate method to measure the expression difference of objects between two different classes [30]. In each TNBC subtype, the CM1 score for each gene in the subtype was calculated by:(1)$$\mathrm{CM}1_{i}=\frac{{\bar{x}}_{i}-{\bar{y}}_{i}}{1+\left(\max\left\{y_{i}\right\}-\min\left\{y_{i}\right\}\right)}$$
where ${\bar{x}}_{i}$ is the average expression value of the gene i in the subtype, ${\bar{y}}_{i}$ is the average expression value of the gene i in other subtypes; max{$y_{i}$} and min{$y_{i}$} represent the maximum and minimum expression values of gene i in the other subtypes.

Then, the CM1 scores for all genes were calculated and arranged in descending order. The most important eight genes (the top and bottom four genes correspond to the up- or down-regulated ones, respectively) were then selected from the ranking list, as candidate genes for the subtype. When the candidate genes for four TNBC subtypes were determined, the overlapping genes in any two candidate gene lists were removed, and the corresponding TSPSigs were generated.

### 2.5. Function Enrichment Analysis

We performed an enrichment analysis for TSPSigs using the R package “clusterProfiler”, based on the canonical pathway gene set collections from the MSigDB database (https://www.gsea-msigdb.org/gsea/msigdb/ (accessed on 1 December 2022)) (*p* value < 0.05) [31].

### 2.6. Representative Evaluation of TSPSigs

To determine whether TSPSigs were representative, we compared the differences in the gene expression, the information entropy, and the inter-gene expression correlation between TSPSigs and the background genes (generated by subtracting TSPSigs from 12,625 genes detected by scRNA-seq). 

### 2.7. Prognostic Evaluation of TSPSigs

We evaluated the prognostic powers of TSPSigs, using data from TCGA, METABRIC, GSE58812, and GSE96058. For each TNBC subtype, patients in validation cohorts were separated into high- and low-risk groups by the cutoff point defined by the “survminer” package [32]. A Kaplan-Meier (K-M) survival curve and log-rank test were then used to evaluate the difference in the overall survival (OS) between these two groups. For each TSPSig, to detect whether it could be an independent prognostic factor, univariate and multivariate Cox regression analyses were carried out with TSPSig and several clinical features (age, stage, chemotherapy, tumor size, and hormone therapy) using the R package “survival”. Then, the nomogram model was constructed for predicting the OS in patients with this TNBC subtype [33]. The calibration curve and time-dependent receiver operating characteristic (ROC) curve analyses were performed to validate the accuracy of each nomogram model for predicting the OS of the TNBC patients at 3- and 5-years. In addition, the prognostic analysis of the TNBC patients who had received chemotherapy in TCGA, METABRIC, and GSE96058, was performed to estimate the predictive powers of TSPSigs. Similarly, a survival analysis of the disease-free interval (DFI), relapse-free survival (RFS), and metastasis-free survival (MFS) in the TCGA, METABRIC, and GSE58812 datasets were carried out, respectively.

### 2.8. Analysis of the TSPSigs Expressions Correlation with the Drug Sensitivities

To explore the effect of TSPSig on the therapeutic responses in each TNBC subtype, we investigated the relations between the TSPSig expression and drug sensitivity. We downloaded the expression and drug sensitivity data (the half maximal inhibitory concentration (IC50)) for the breast cancer cell lines from the Genomics of Drug Sensitivity in Cancer (GDSC) (https://www.cancerrxgene.org/ (accessed on 17 November 2021)) and extracted 26 TNBC cell lines, which were then classified by the TNBCtype-4 subtyping approach. For each TNBC subtype, the TNBC cell lines were divided into high- and low-expression groups, based on the median expression value of TSPSig, and the differences in the IC50 values of the compounds between these two groups were compared. The Spearman correlation coefficients were used to evaluate the correlations between the TSPSig expression levels and the drug sensitivity to 367 compounds (*p* < 0.05 was considered significantly related).

## 3. Results

### 3.1. Identification of TSPSigs, Based on the scRNA-Seq Data

With the advantages of the scRNA-seq technology in enabling the accurate characterization of the cell properties from complex tissues, we carried out TNBC subtype analyses at a single-cell resolution and aimed to identify TSPSigs for further uncovering the potential pathogenic mechanisms of them. So GSE176078, which has 10 TNBC patients associated with 42,512 cells was selected and used. To eliminate the influence of non-disease cells, 10,836 cancer epithelial cells with a “pure” disease state were extracted from their original cell annotation file (Figure 2A). We removed the low-quality cells and protein-coding genes with a too low expression, 10,701 cells and 12,625 genes finally remained. Following the classification of the cancer epithelial cells by the TNBCtype-4 subtyping approach, the numbers of cells of BL1-, BL2-, LAR-, and M-subtypes were 996, 614, 842, and 84, respectively. The application of the CM1 score method, 4, 6, 6, and 6 genes were separately discovered and considered as the BL1-, BL2-, LAR-, and M-TSPSig (Figure 2B and Appendix A). We found that eight out of the 22 genes of four TSPSigs were TNBC known disease genes and they were recorded in DisGeNET (https://www.disgenet.org/ (accessed on 18 October 2021)), including CD24, AR, EPCAM, NFIB, S100A4, IDH2, LGALS1, and AZGP1 (Figure 2C). Among these, CD24 was found to be a promising treatment target for TNBC and its overexpression was markedly associated with a shorter OS in TNBC [34,35]. Several reports found that AR was highly expressed in the LAR-subtype, compared with other subtypes and demonstrated the prognostic worth of AR in TNBC [12,36]. As a tumor-associated antigen, the EPCAM overexpression was in connection with a poorer prognosis in most TNBC tumors [37,38]. NFIB was known for having a higher expression in TNBC and could be served as a potential therapeutic target in the TNBC patients [39,40]. It has been found that S100A4 was related to the TNBC cell motility, invasion, and metastasis [41]. Mutations of the IDH family gene-IDH1 and IDH2 were most frequently detected in TNBC patients during the analysis of the discordance between the immunohistochemistry (IHC)-based surrogate subtyping and the PAM50 intrinsic subtypes [42]. Galectin-1, encoded by the LGALS1 gene, was found to be significantly up-regulated in the TNBC patients and was regarded as a potential TNBC-specific cell surface marker [43]. AZGP1 was overexpressed in TNBC and might be used as a potential marker to discriminate TNBC from other non-TNBC tumors [44]. These findings further demonstrated that the TSPSig genes were closely related to TNBC.

Functional enrichment analysis was then performed, and the results revealed the functional heterogeneity among TSPSigs and their close associations with cancer development (Appendix A). Among that, BL1-TSPSig was mainly enriched in the “RHOB GTPase cycle”, “cell adhesion molecules (CAMs)”, and the “CXCR4-mediated signaling events” pathways. BL2-TSPSig was mainly enriched in the “regulation of TLR by endogenous ligand”, “endogenous TLR signaling”, and “RHO GTPases activate NADPH oxidases” pathways. LAR-TSPSig was enriched in the “regulation of the androgen receptor activity”, “RHO GTPase effectors”, and “TNF-alpha signaling” pathways. M-TSPSig was mainly correlated with the “mitochondrial biogenesis”, “pyruvate metabolism and citric Acid (TCA) cycle”, and the “tyrosine metabolism” pathways, etc.

### 3.2. TSPSigs Were More Representative

We then compared the differences between TSPSigs and the background genes in the gene expression, information entropy, and inter-gene expression correlation, and found that most TSPSigs were significantly higher than the background genes in the above three aspects, indicating that TSPSigs were representative in their corresponding subtypes (Figure 2D,E). 

### 3.3. Evaluation of the TSPSigs’ Prognosis in Four Validation Cohorts

To evaluate the prognostic powers of TSPSigs, we performed a prognosis analysis in four TNBC validation datasets, including TCGA (*n* = 123), METABRIC (*n* = 299), GSE96058 (*n* = 151), and GSE58812 (*n* = 107) (Table 1). The results showed that BL1-TSPSig could efficiently stratify the BL1-subtype patients with an OS into the high-risk group (*n* = 27 and 13, respectively) and low-risk group (*n* = 2 and 92) in TCGA (HR = 0.15, *p* = 0.026) and METABRIC (HR = 3.15, *p* = 0.001) (Figure 3A). BL2-TSPSig was able to successfully stratify the BL2-subtype patients into high- and low-risk groups (*n* = 3 and 16) in TCGA (HR = 11, *p* = 0.056) (Figure 3B). LAR-TSPSig could separately classify the LAR-subtype patients into the high-risk group (*n* = 20, 4 and 6) and low-risk group (*n* = 40, 8, and 16) with significantly different OS in METABRIC (HR = 1.92, *p* = 0.042), GSE58812 (HR = 10.64, *p* = 0.041), and GSE96058 (HR = 4.75, *p* = 0.048) (Figure 3C). M-TSPSig also had a good efficiency in classifying the M-subtype patients into high- and low-risk groups (*n* = 8 and 14) in GSE58812 (HR = 4.92, *p* = 0.012) (Figure 3D).

What’s more, we also explored the effectiveness of each TSPSig in predicting other clinical outcomes with DFI, RFS, and MFS in TCGA, METABRIC, and GSE58812, respectively. The results showed that BL1-TSPSig could effectively predict DFI (HR = 0.11, *p* = 0.057) and RFS (HR = 3.18, *p* = 0.002) of the BL1-subtype patients in TCGA and METABRIC, respectively (Appendix A). BL2-TSPSig could successfully predict DFI (HR = 0.03, *p* = 0.001) of the BL2-subtype patients in TCGA (Appendix A). LAR-TSPSig had good predictive powers in predicting MFS (HR = 10.22, *p* = 0.044) of the LAR-subtype patients in GSE58812 (Appendix A). M-TSPSig could successfully predict MFS (HR = 5.48, *p* = 0.008) of the M-subtype patients in GSE58812 (Appendix A). Taken together, most TSPSigs exhibited a good prognostic efficacy in predicting the OS and other clinical outcomes of the TNBC subtype patients.

### 3.4. Independent Prognostic Factor Assessment and the Nomogram Construction

To further investigate whether each TSPSig was a prognostic factor independent of other clinical factors in the validation datasets, the univariate and multivariate Cox regression analyses were performed (Table 2). Due to the small sample size (<50) of the TNBC subtypes in the TCGA and GEO datasets, only the METABRIC dataset with 299 TNBC patient samples could be used for the analysis. We found that BL1-TSPSig (HR = 2.692, *p* = 0.020) was an independent prognostic factor for patients with the BL1-subtype. Stage (HR = 3.293, *p* = 0.037) was an independent prognostic factor for patients with the BL2-subtype. In the LAR-subtype, LAR-TSPSig (HR = 2.369, *p* = 0.028) and tumor size (HR = 1.021, *p* = 0.018) were two independent prognostic factors. While, tumor size (HR = 1.028, *p* = 0.035) and stage (HR = 9.161, *p* = 0.014) were two independent prognostic factors for patients with the M-subtype.

Based on the multivariate Cox regression analysis, we then constructed a comprehensive nomogram for each TNBC subtype to predict the overall survival of the TNBC patients at the 3- and 5-year follow-ups (Figure 4A). A calibration curve and ROC curve were used to measure the predictive accuracy of the nomogram. The calibration curve of each TNBC subtype showed a good agreement between the predicted and actual OS (Figure 4B). For the BL1-, BL2-, LAR-, and M-subtypes, the area under the ROC (AUC) values of 3- and 5-years were 0.708 and 0.653, 0.655 and 0.705, 0.779 and 0.787, and 0.790 and 0.771, respectively, confirming that the nomogram models had a good capacity in predicting the OS (Figure 4C).

Similarly to the OS analysis, we also performed the above analysis for the RFS in the METABRIC dataset, and the results showed that BL1-TSPSig and LAR-TSPSig were also independent prognostic factors in predicting the RFS, and the nomogram performed well in predicting the RFS in the corresponding TNBC subtype patients, which was consistent with the result of predicting the OS (Appendix A).

### 3.5. The Predictive Powers of TSPSigs in Patients with Chemotherapy

Usually, TNBC patients are more sensitive to chemotherapy, but their tumors tend to have a high risk for the disease progression [45,46,47], so we further examined the predictive powers of each TSPSig for patients with chemotherapy in TCGA, METABRIC, and GSE96058, which had 83, 157, and 104 patients, respectively (Appendix A). Through the analysis, we found that BL1-TSPSig could perfectly classify the BL1-subtype patients into high- (*n* = 8 and 19) and low-risk groups (*n* = 59 and 19) with a markedly different OS in METABRIC (HR = 3.69, *p* = 0.002) and GSE96058 (HR = 0.11, *p* = 0.037) (Figure 5A). Similarly, M-TSPSig had a good performance in stratifying the M-subtype patients with chemotherapy into high- and low-risk groups (*n* = 6 and 15) in METABRIC (HR = 0.22, *p* = 0.038). Yet, BL2-TSPSig and LAR-TSPSig were less able to distinguish patients with chemotherapy into high- and low-risk groups, significantly, whose *p* values ranged from 0.068 to 0.087 in TCGA, METABRIC, and GSE96058.

### 3.6. TSPSigs Expressions Were Associated with Drug Sensitivities

To further study the links between TSPSigs and the drug activities on the TNBC cell lines, we obtained the gene expression and drug sensitivity data (IC50) of the TNBC cell lines from GDSC. These cell lines were then classified into TNBCtype-4 subtypes, including 6 BL1-, 8 BL2-, 4 LAR-, and 6 M-subtypes (Appendix A). For each subtype, the high- and low-expression groups were determined by their corresponding TSPSig and the differences in the drug sensitivity between these two groups were evaluated. We found that the IC50 values of seven compounds (Avagacestat, Pazopanib, BX-912, GSK1070916, JQ1, KU-55933, and OSI-027) in the BL2-TSPSig high-expression group were significantly lower than in the low-expression group (*p* = 0.029), which suggested that the BL2-subtype cell lines with a higher expression of BL2-TSPSig, were more sensitive to these seven drugs (Figure 5B).

Next, a Spearman correlation analysis was performed to evaluate the correlations between four TSPSigs expression levels and drug sensitivities. We found that the expressions of BL1-, BL2-, and M-TSPSig were significantly correlated (*p* < 0.05) with the sensitivities of multiple drugs in their corresponding TNBC subtype cell lines (Figure 5C). Among them, higher expression levels of BL1-TSPSig were associated with the increased sensitivity of the BL1-subtype cell lines to PF-4708671 and Linifanib but led to an increased resistance to SN-38 and PARP_0108. Higher expression levels of BL2-TSPSig could increase the sensitivity of the BL2-subtype cell lines to Avagacestat and Alisertib. What’s more, the M-subtype cell lines with higher M-TSPSig expression levels were more sensitive to AZD7762, MIM1 and LCL161, and so on, but resistive to several compounds, such as Pictilisib, BPTES, and GSK269962A.

## 4. Discussion

To our knowledge, studies on systematically identifying the TNBC subtype-specific prognosis gene markers are still very lacking. Depending on the TNBC subtype-related cells with a “pure” disease state extracted from the scRNA-seq data, we developed a single-cell RNA-seq-based bioinformatics approach to identify TSPSigs and tried to capture TSPSigs with real pathogenic functions in their respective subtypes. Through an analysis of the TNBC scRNA-seq data of GSE176078, TSPSigs were identified, and we found some TSPSigs were known disease genes and had a good capability in predicting patient survival and effect on the drug response in the validation datasets. Therefore, we thought that these TSPSigs would help to further open up new ideas for TNBC research and benefit the personalized treatment.

Since distinct TNBC subtypes are biologically diverse and have a highly intertumoral heterogeneity, it is very important and necessitated to classify its subtypes more accurately in most research studies, so we applied the TNBCtype-4 subtyping approach, one of the most widely used gene expression-based molecular classifications, to stratify all TNBC patient cells or samples, no matter the scRNA-seq, RNA-seq and gene expression data from GEO, TCGA, and METABRIC. We considered that its precise stratification in the TNBC subtypes enables us to conduct individualized studies for patients in each subtype, facilitating the identification of the subtype-specific prognosis signatures. To further construct TSPSigs, we integrated the ranking feature method named the CM1 score, whose original function is suitable for identifying specific biomarkers for individual subtypes in our approach. Indeed, we found that TSPSigs captured by the CM1 score method were more representative and they were significantly higher than background genes in the gene expression, information entropy, and inter-gene expression correlation. In these TSPSigs, except the aforementioned TNBC known disease-related genes described in the “Results” section, we also found that some other genes of TSPSigs might serve as potential therapeutic and prognostic gene targets in TNBC, which reinforced the associations between TSPSigs and TNBC. For instance, RHOB was found to be critical for the TNBC cell migration and the RHOB inhibitors could be conducive to the improved therapy response of the TNBC patients with brain metastases [48]. The activity of UCHL1 was significantly increased in the TNBC cell lines when compared to the non-TNBC cell lines and it might be a potential therapeutic target for TNBC [49,50]. Recent research found that S100A9 was significantly highly expressed in the TNBC subtype, which was associated with poorer clinical outcomes, and it could be used as a prognostic indicator of TNBC [51]. Zhou et al. demonstrated that SPDEF might be closely associated with the development and prognosis of TNBC [52]. By studying the relationship between the overexpression of MIF and the growth and metastasis of TNBC, Charan et al. found that using MIF inhibitors may inhibit the progression and metastasis of TNBC [53]. Summarily, combined with the evaluation results provided above, we believed that these TSPSigs could be used as potential prognostic markers to guide the prognosis treatment of TNBC patients. Interestingly, we found that higher expression levels of BL1- and M-TSPSig led to the increased resistance of the corresponding subtype cell lines to some drugs, whose drug targets were, respectively, TOP1, PI3K, GLS, EPHB4, the PARP family gene (PARP1, PARP2 and PARP6), and the ROCK family gene (ROCK1 and ROCK2). With the increase of the TSPSigs expression levels, the drug targets and pathways might be affected, leading to drug resistance, but the specific reason of TSPSigs in mediating drug activities may need further exploration in the future.

There were some shortcomings in our approach. Firstly, the TNBCtype-4 approach could filter a large number of samples that did not meet the classification threshold criteria during the subtyping, so the small sample size of each subtype in TCGA and GEO potentially hampered the evaluations of TSPSigs for judging them as independent prognostic factors; Secondly, although some known disease genes of TNBC existed and were recorded in DisGeNET, the associations between the TNBC subtype and the known disease genes were rarely reported, which makes it difficult to carry out a more comprehensive TSPSig assessment, therefore, more in-depth explorations were needed to be conducted in future work.

## 5. Conclusions

We proposed a single-cell RNA-seq-based approach to identify TSPSigs in specific TNBC subtypes and analyzed them from multiple aspects. The analysis results demonstrated that TSPSigs were closely correlated with the TNBC tumor development and had a good ability to distinguish the distinct TNBC subtype patients with different clinical prognostic outcomes. We expected that the systematic study of TSPSigs at a single cell resolution will help provide a novel direction for exploring the subtype-related gene biomarkers for the TNBC diagnosis and treatment, which will benefit the personalized treatment strategies for the TNBC subtype patients.

## Figures and Tables

**Figure 1 cells-12-00367-f001:**
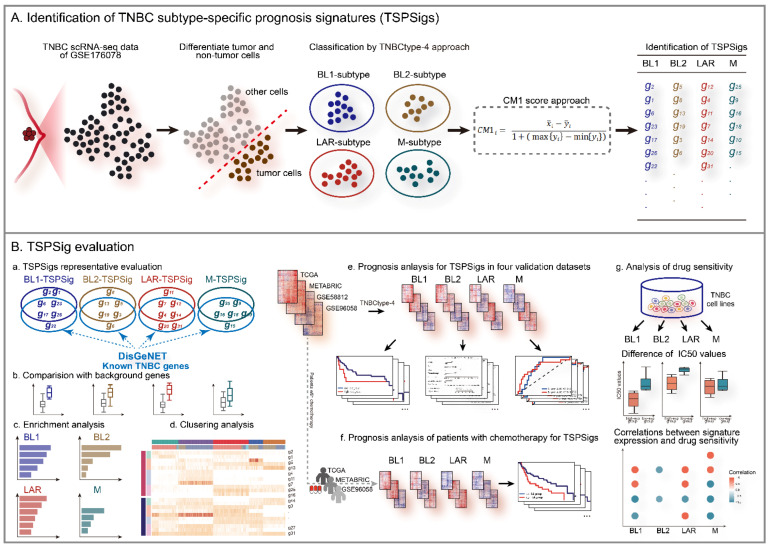
The workflow of this study.

**Figure 2 cells-12-00367-f002:**
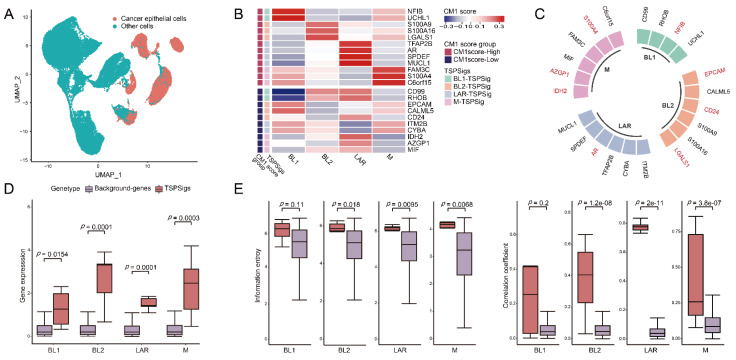
Identification of TSPSigs. (**A**) UMAP plot for the TNBC cancer epithelial cells and other cells in GSE176078. (**B**) Heatmap showed the CM1 scores of genes in TSPSigs. Annotated row: (**1**) CM1 score group: positive- and negative-CM1 scores were colored in red and purple, respectively, and (**2**) TSPSigs. (**C**) Plot for the known TNBC disease genes (colored in red) in each TSPSig. Differences between TSPSigs and the background genes in (**D**) the gene expression, (**E**) information entropy, and the inter-gene expression correlation.

**Figure 3 cells-12-00367-f003:**
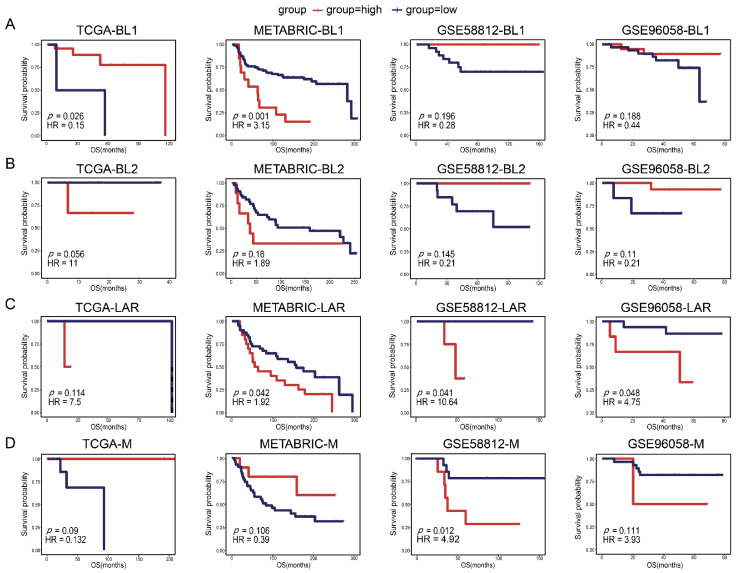
Kaplan-Meier survival curves of the OS between the high-risk and low-risk patients with each TNBC subtype in four validation datasets (TCGA, METABRIC, GSE58812, and GSE96058). (**A**) BL1-subtype, (**B**) BL2-subtype, (**C**) LAR-subtype, (**D**) M-subtype.

**Figure 4 cells-12-00367-f004:**
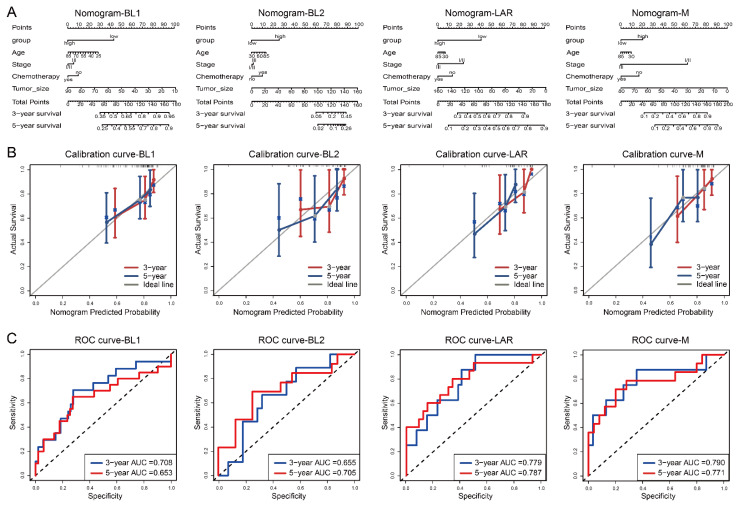
Nomogram, calibration curve and the AUC of the nomogram model, based on the ROC curve used to predict the overall survival time of patients with each subtype in the METABRIC dataset. (**A**) Nomogram, (**B**) calibration curve, (**C**) ROC curve.

**Figure 5 cells-12-00367-f005:**
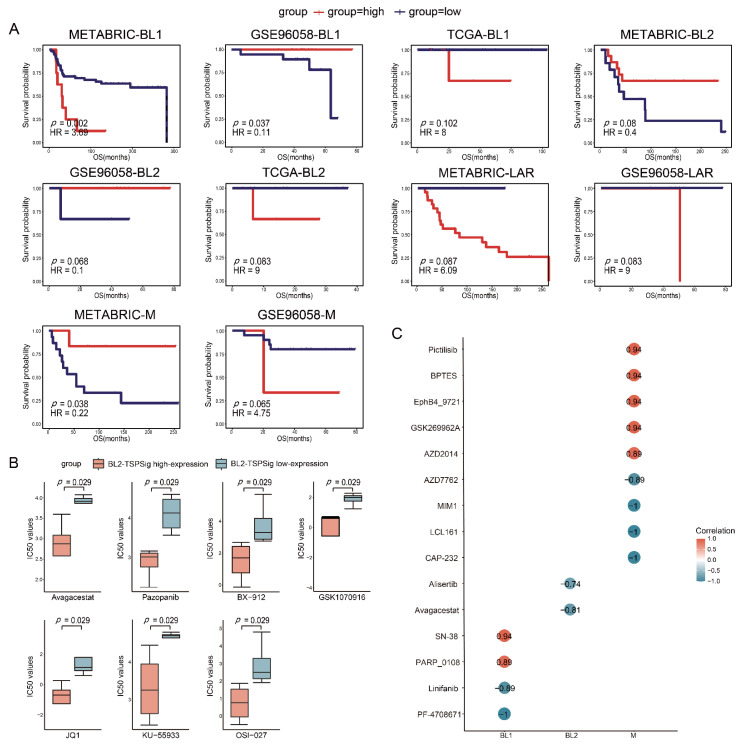
Estimation of the predictive powers of TSPSigs and the relationships between the TSPSigs expressions and drug sensitivities. (**A**) Kaplan-Meier survival curves of the OS between high-risk and low-risk patients with chemotherapy of each TNBC subtype in three validation datasets (METABRIC, GSE96058, and TCGA). (**B**) Differences in the IC50 values of the BL2-subtype cell lines treated with seven drug components between the BL2-TSPSig high- and low-expression groups. (**C**) Correlations between each TSPSig expression level and drug sensitivity (IC50). A positive (or negative) correlation means that the high expression of TSPSig was resistant (or sensitive) to the drug.

**Table 1 cells-12-00367-t001:** Summary of the TNBC subtype patients/epithelial cells in the datasets used in this study.

Series	Platforms	No. of Samples/Cells	BL1	BL2	LAR	M	Others
GSE176078	Illumina NextSeq 500 (GPL18573)	10,836 cells	996	614	842	84	8300
TCGA	Illumina HiSeq 2000	123 samples	29	19	13	29	33
METABRIC	Illumina HT-12 v3	299 samples	105	55	60	54	25
GSE58812	Affymetrix Human Genome U133 Plus 2.0 Array (GPL570)	107 samples	30	17	12	22	26
GSE96058	Illumina HiSeq 2000 (GPL11154) Illumina NextSeq 500 (GPL18573)	151 samples	49	21	22	32	27

**Table 2 cells-12-00367-t002:** Univariate and multivariate Cox regression analyses of the OS in each TNBC subtype.

	Univariate Analysis			Multivariate Analysis		
Variables	HR	95% CI	*p*-Value	HR	95% CI	*p*-Value
BL1-subtype						
BL1-TSPSig (high vs. low)	2.716	1.258–5.865	0.011 *	2.692	1.171–6.188	0.020 *
Chemotherapy (yes vs. no)	1.027	0.524–2.010	0.939	1.567	0.643–3.819	0.323
Age	1.021	0.994–1.049	0.126	1.018	0.984–1.053	0.302
Tumor_size	1.031	1.006–1.056	0.014 *	1.031	0.999–1.064	0.062 .
Stage (III vs. I/II)	1.310	0.314–5.472	0.711	0.597	0.107–3.342	0.557
BL2-subtype						
BL2-TSPSig (high vs. low)	2.006	0.656–6.139	0.222	1.330	0.208–8.525	0.763
Chemotherapy (yes vs. no)	0.609	0.241–1.539	0.295	0.455	0.123–1.682	0.238
Age	1.035	1.000–1.074	0.067 .	1.028	0.981–1.077	0.247
Tumor_size	1.023	1.003–1.045	0.025 *	1.017	0.989–1.045	0.237
Stage (III vs. I/II)	3.400	1.320–8.759	0.011 *	3.293	1.075–10.093	0.037 *
LAR-subtype						
LAR-TSPSig (high vs. low)	2.361	1.129–4.938	0.023 *	2.369	1.100–5.105	0.028 *
Chemotherapy (yes vs. no)	1.396	0.680–2.863	0.364	1.404	0.561–3.515	0.468
Age	1.015	0.982–1.050	0.376	1.008	0.968–1.051	0.694
Tumor_size	1.023	1.007–1.039	0.006 **	1.021	1.004–1.039	0.018 *
Stage (III vs. I/II)	2.130	0.805–5.627	0.128	1.049	0.335–3.283	0.934
M-subtype						
M-TSPSig (high vs. low)	0.497	0.147–1.674	0.259	0.501	0.143–1.759	0.281
Chemotherapy (yes vs. no)	1.195	0.515–2.775	0.678	1.283	0.372–4.424	0.693
Age	0.991	0.957–1.027	0.624	1.001	0.954–1.050	0.970
Tumor_size	1.025	1.000–1.051	0.046 *	1.028	1.002–1.055	0.035 *
Stage (III vs. I/II)	10.38	2.079–51.830	0.004 **	9.161	1.558–53.863	0.014 *

TSPSig, the TNBC subtype-specific prognosis signature; **, *p* < 0.01; *, 0.01 ≤ *p* < 0.05; ., 0.05 ≤ *p* < 0.1.

## Data Availability

All data are available in the main text or in the Appendix A.

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
