# Peer review of "Systemically Identifying Triple-Negative Breast Cancer Subtype-Specific Prognosis Signatures, Based on Single-Cell RNA-Seq Data"

_cells, 2023, doi:10.3390/cells12030367_

Round 1

Reviewer 1 Report

The article is focus on the relationship between TSPSigs and TNBC subtypes. The results showed that  1.TSPSigs in two TNBC subtypes-BL1 and LAR were two independent prognostic factors; 2. TSPSigs expressions and drug sensitivities had significant associations. The content of the article is well organized, rich in content, deep in database mining, and reliable in data. However, if receive this article, it needed to make major revision.

1.This article mainly focuses on bioinformatics analysis, lacking in vitro and in vivo validation, and further validation is recommended.

2.There is a lot of gene related information related to drug resistance. Can the author discuss the possibility, possible pathways and reasons of this information?

3.There are some errors in the Table 2 about the use of "*".

Reviewer 2 Report

This manuscript by Kaiyuan Xing and colleagues studied Systematic Identification of Triple-negative Breast Cancer Subtype-Specific Prognosis Signatures Using Single-Cell RNA-Sequencing Data.

In this study the authors presented the following:

The authors identified TSPSigs in certain TNBC subtypes using a single-cell RNA-seq technique. The results of the analysis demonstrated that TSPSigs were highly correlated with the development of TNBC tumors and distinguished TNBC subtype patients with distinct clinical prognoses.

This is an interesting study, but the following concerns should be addressed during revision:

1.     Fig2D, in gene expression analysis, what do you mean by "background genes"?

2.     In this analysis, what did you use as a control for TNBC?

3.     What are TSPSigs expression levels in normal cells in comparison with TNBC cells?

4.     Did you validate your findings?

5.     Fig 5C, Did you check the expression levels of TSPSigs in TNBC cell lines in the presence of seven compounds (Avagacestat, Pazopanib, BX-912, GSK-1070916, JQ1, KU-55933, and 324 OSI-027) in comparison to untreated cells?

6.     Please write your conclusion based on what you found at the end of each results section to help the reader understand.

Reviewer 3 Report

The study of Xing et al. provided an important investigation on identifying new TNBC subtype-specific prognosis signatures using single cell RNA-seq data from cancer patients. The prognostic power of identified cell-type specific signatures could be validated in other public breast cancer datasets. The authors also proved the correlations between TSPSigs expressions and drug sensitivities. All above results indicate the potentials of TSPSigs on the prognostic determination and therapeutic plans for TNBC patients.

There are a few issues the authors should address:

1. Authors claimed they selected 8 top genes in each cell type but finally only identified no more than 6 genes per cell type. It is unclear why authors only considered a small number of genes. The following results in validation steps (section 3.3) clearly showed that the performances of some TSPSigs were not so good in other datasets. In discussion, authors argued that the small sample sizes in TCGA and GEO affecting the evaluations. But more genes can be involved in constructing TSPSigs to increase its universality for diverse datasets.

2. For function enrichment analyses, it is expected that authors used 24 TSPSigs genes for analysis. But the “Size” in Supplementary Table S2 is much larger than that. Authors should revise the corresponding paragraph in methods for details. Further, the results only showed GO analysis in BL2 and LAR and KEGG in M. Authors need also provide the explanations for the lack of other cell types.

3. 8 genes were found in DisGeNET and it is great that authors provided references with their known effects in TNBC. However, the connections between cell types and genes were poorly investigated, except the gene AR. Authors may provide more information here, e.g., Honeth, et al., 2008 for CD24- in basal like breast tumor.

4. The consensus clusters are not consistent with 4 cell types. It could be observed that LAR and BL2 were always mixed, while M was not separated from other cell types. Maybe authors can perform a unsupervised cell type classification using UMAP and compare it with TSPSigs.

Minor issues:

1. In the figure 1A, authors may re-organized second figure. It is confused that the TNBCtype-4 cells were analyzed using “other cells” instead of “tumor cells”.

2. In page 7 line 227, authors may clarify which genes are “background” genes. Are those genes not included in TSPsigs defined as bbackground genes?

3. Figure 5B should be re-plotted or formatted as table. It is confusing why PARP_0108 and Avagacestat are in the same lines, and why AZD7762 and GSK269962A are not grouped with other drugs at same correlation trends. In addition, Figure 5C is mentioned earlier than 5B.

4. In page 12 line 344, the word “most” is inappropriate as only 8 of 22 genes are found in DisGeNET.

Round 2

Reviewer 1 Report

The revised draft has reached the accept standard, so it can be accepted.

Reviewer 3 Report

No more comments